# Comparative Study of Three Mixing Methods in Fusion Technique for Determining Major and Minor Elements Using Wavelength Dispersive X-ray Fluorescence Spectroscopy

**DOI:** 10.3390/s20185325

**Published:** 2020-09-17

**Authors:** Dan-Ping Zhang, Ding-Shuai Xue, Yan-Hong Liu, Bo Wan, Qian Guo, Ju-Jie Guo

**Affiliations:** 1State Key Laboratory of Lithospheric Evolution, Institute of Geology and Geophysics, Chinese Academy of Sciences, Beijing 100029, China; zdanp@mail.iggcas.ac.cn; 2Innovation Academy for Earth Science, Chinese Academy of Sciences, Beijing 100029, China; 3Institute of Geology and Geophysics, Chinese Academy of Sciences, Beijing 100029, China; liuyanhong@mail.iggcas.ac.cn (Y.-H.L.); wanbo@mail.iggcas.ac.cn (B.W.); guoqian@mail.iggcas.ac.cn (Q.G.); guojujie@mail.iggcas.ac.cn (J.-J.G.)

**Keywords:** wavelength dispersive X-ray fluorescence spectroscopy, sample preparation procedure, fused glass disc, shaker cup method

## Abstract

Accurate analysis using a simple and rapid procedure is always the most important pursuit of analytical chemists. In this study, a new sample preparation procedure, namely the shaker cup (SH) method, was designed and compared with two sample preparation procedures, commonly used in the laboratory, from three aspects: homogeneity of the sample–flux mixture, potential for sample contamination, and sample preparation time. For the three methods, a set of 54 certified reference materials (CRMs) was used to establish the calibration curves, while another set of 19 CRMs was measured to validate the results. In the calibration procedures, the matrix effects were corrected using the theoretical alpha coefficient method combined with the experimental coefficient method. The data of the major oxides (SiO_2_, TiO_2_, Al_2_O_3_, TFe_2_O_3_, MnO, MgO, CaO, Na_2_O, K_2_O, and P_2_O_5_) and minor elements (Cr, Cu, Ba, Ni, Sr, V, Zr, and Zn) obtained by wavelength dispersive X-ray fluorescence spectroscopy (WD-XRF) were compared using two derivative equations based on the findings by Laurence Whitty-Léveillé. The results revealed that the WD-XRF measured values using the SH method best agreed with the values recommended in the literature.

## 1. Introduction

Wavelength dispersive X-ray fluorescence spectrometry (WD-XRF) is a widely used qualitative and quantitative analytical method, which has been applied in many fields including geology, geochemistry, mining, metallurgy, non-ferrous metals, building materials, environmental protection, and commodity inspection [1,2,3,4,5,6,7,8]. In geology and geochemistry, WD-XRF is mainly used to analyze the elemental composition of rocks, which can be employed for the exploration of mineral deposits and classification of the whole rock diversity [9,10]. Atomic absorption spectrometry (AAS), inductively coupled plasma atomic emission spectrometry (ICP-AES), and inductively coupled plasma mass spectrometry (ICP-MS) are also used for the analysis of major, minor, and trace elements in rocks; however, these instrumental methods require tedious wet chemical pretreatment. Thus, burdensome labor can easily cause accidental contamination and uncertainty of analysis, while many reagents that are toxic to the operator are consumed [11]. In contrast, XRF presents numerous advantages including simple sample preparation, no environmental pollution, and little training for personnel. This technique can also detect over a wide range of elements and can be employed to analyze multiple elements simultaneously. Moreover, XRF instruments have long-term stability enabling sample analysis with high precision and reproducibility. To take full advantage of these features, however, one needs to pay particular attention to the sample preparation process. This is because such processes have a very high potential for introducing systematic errors into the XRF analysis.

The most common sample preparation methods used for the analysis of rock samples by conventional WD-XRF instrumentation are pressed powder pellets and fused glass discs [12,13]. Fused glass discs, which is the most extensively used analytical technique in the laboratory, is excellent for analyzing major and minor elements as it reduces matrix effects, eliminates particle size effects, and provides a homogeneous specimen [14,15]. Many studies to determine the major and minor elements using WD-XRF have been reported in the literature [9,10,16,17,18]. The results have revealed that 1:10 dilution glass discs are best suited for major and minor element analysis as this ratio accounts for the sample, reagent dosage, analysis signal strength, and simplicity and feasibility of the operation procedure. Preferably, the sample should be homogeneously mixed with flux before it is fused to a glass disc. Thus, different sample preparation procedures for making fused glass discs have been reported. For example, Shintaro Ichikawa and Toshihiro Nakamura mixed the sample and flux on paraffin paper using a bamboo spatula and subsequently placed the mixture into a Pt–Au crucible for fusing [19]. Masatsugu Ogasawara et al. also used weighing paper and mixed the sample and flux thoroughly with a spatula. Additionally, they suggested another alternative in which the sample and flux powder are transferred to an agate mortar, mixed thoroughly, and subsequently transferred into a Pt–Au crucible for fusing [20]. Atsushi Goto and Yoshiyuki Tatsumi reported a mixing procedure in which the powder rock sample was transferred from the weighing paper onto the flux and the two were then mixed by raising the sides of the weighing paper; however this technique is difficult to operate and the sample can be easily spattered [16]. Kenichiro Tani et al. used a vibration touch-mixer to mix the sample and flux, which were both directly weighed into the same Pt–Au crucible [21]. Yuji Orihashi and Takafumi Hirata also reported mixing using a touch-mixer [22]. However, heterogeneous glass discs made by this method have only been successfully used to mix an aqueous solution in which convection occurs effectively, which is not observed in solid powders. W. B. Stern reported a method in which an ignited powder sample was carefully mixed in an agate mortar with dried Li_2_B_4_O_7_ and subsequently transferred into a Pt–Au crucible for fusing [23]. However, these mixing methods were not described in enough detail. Moreover, many authors such as Toru Yamasaki [24] and Atsushi Kamei [25] did not even report the mixing method employed.

In view of the XRF analysis error mainly originating from the sample preparation procedure, this step is extremely important for the whole XRF analytical process [26]. Therefore, it is essential to explore a simple and effective way to prepare a homogeneous glass disc, prior to XRF analysis, from which more accurate and precise analytical results can be obtained.

In our lab, two sample preparation procedures, namely the grinding (GR) and stirring rod (ST) methods, have been mainly used to prepare fused glass discs. The GR method comprises mixing of the sample and flux in an agate mortar and subsequent transfer of the afforded mixture to a Pt–Au crucible with a spoon and brush for the next fusing step; this method was also used by Stern [23]. The advantage of this method is that the sample and flux can be mixed very evenly, whereas its main disadvantage is the ease of contamination and sample loss during the transfer process. The ST method comprises the direct weighing of the sample and flux into a Pt–Au crucible, mixing with a glass rod, and subsequent fusion of the resultant mixture in a fusion machine. The advantage of this method is that it is a simpler process with a lower risk of contamination and sample loss during the transfer process. On the other hand, its main disadvantage is insufficient mixing of the sample and flux.

Herein, we propose an improved sample preparation procedure, namely the shaker cup (SH) method, inspired by the protein shaker cup used in bodybuilding. This method combines the advantages of the GR and ST methods to conveniently mix the sample and flux homogenously in a cheap shaker cup. Moreover, this method allows easy transfer of the afforded mixture to a Pt–Au crucible. To evaluate the quality of the data obtained from the different glass disc procedures, calibration curves were established and major and minor elements of 19 certified reference materials (CRMs) were examined. The operational processes, lower limits of detection (LLD), accuracy, and precision were compared in detail. The results revealed that of the three investigated methods, the SH method is the most suitable for creating fused glass discs.

## 2. Experimental

### 2.1. Instrumentation

An electric drying oven (WGL-125B, Tianjin Taisite Instrument Co., Ltd., Tianjin, China) was used to dry the samples and flux, while a muffle furnace (SX-GO4133, Tianjin Zhonghuan Test Electrical Furnace Co., Ltd., Tianjin, China) was used to burn the samples. The samples and flux were weighed with an electronic balance (ME104E, Mettler-Toledo, Zurich, Switzerland). An ultrasonic cleaning device (Elma E100H, Elmasonic, Singen, Germany) was used to clean the Pt–Au crucibles and molds, while a six-position automatic fusion device (Claisse Fluxy, Corporation Scientific Claisse Inc., Quebec, Canada) was used to create the fused glass discs. Imaging studies were performed on an energy dispersive X-ray fluorescence spectroscopy (EDXRF) spectrometer (M4 TORNADO, Bruker Nano GmbH, Berlin, Germany). The X-ray fluorescence spectrometer (AXIOS, PANalytical, Almelo, The Netherlands) was used with a Rh anode X-ray tube and 4 kW excitation power. This is a sequential instrument with a single goniometer-based measuring channel covering the complete elemental measurement range from F to U, in the concentration range from 1.0 ppm to % level determined in vacuum media. SuperQ 4.0 analysis software was used. The spectrometer conditions for ten major and eight minor elements are listed in Table 1.

### 2.2. Reagents

Absolute ethanol (Sinopharm Chemical Reagent Co., Ltd., Shanghai, China) was used for scrubbing the crucibles and agate mortar. Citric acid monohydrate (Sinopharm Chemical Reagent Co., Ltd., Shanghai, China) was used to clean the Pt–Au crucibles and molds, while ammonium bromide (Sinopharm Chemical Reagent Co., Ltd., Shanghai, China) was used as an exfoliation agent. The ultra-pure grade mixed reagent lithium tetraborate:lithium metaborate (67:33; Claisse, Quebec, Canada) was used as an alkali flux to create the fused glass discs.

### 2.3. Reference Materials

Fifty-four CRMs were used to calibrate the spectrometer. We considered the quality of the recommended values for each CRM, required interval of concentration for each element, and previous calibration tests. The source and recommended values for these reference materials were obtained from a series of literature. In the calibration procedure, the matrix effects were corrected using a theoretical alpha coefficient method combined with an empirical coefficient method. Nineteen CRMs were used for the validation of the three sample preparation procedures. The details of the 73 CRMs employed in this study are listed in Appendix A.

### 2.4. Procedure

The procedure used to create the fused glass discs from the rock powder samples is illustrated in Appendix A and the details can be described as follows: first, place the samples and flux into an oven, then dry them at 105 °C for 12 h. Second, take the samples and flux out, then put them into a desiccator to cool for 2 h at 25 °C. Third, weigh 0.6000 g of powered rock sample and 6.000 g of flux. Fourth, mix the weighed sample and flux, then transfer them into a Pt–Au crucible (25 mL; 95% Pt, 5% Au). Fifth, add four drops of 0.12 g mL^−1^ ammonium bromide solution (0.18 mL) into the Pt–Au crucible, then create six fused glass discs simultaneously with the six-position automatic fusion device. The heating time is 19 min at 1050 °C. The cooling time is 285 s. Last, take out the fused glass discs with a suction pen after fusing, then number each one according to the sample name on the non-test surface and put them into a desiccator for subsequent testing.

Notably, most laboratories tend to employ the same drying, cooling, weighing, and numbering steps, while the mixing step tends to be different. Moreover, the adjustable factors involved in automatic melting using the Claisse Ox fusion furnace are beyond the scope of this article and are not elaborated herein. The procedure flow charts are presented in Appendix A and the sample preparation methods are described in detail. For the grinding method, first, wipe 30 5 mL porcelain crucibles and 30 25 mL porcelain crucibles with alcohol. Second, place the 5 mL porcelain crucibles into a muffle furnace and burn at 1000 °C for 30 min, then take them out and put them into a dryer to cool for 1 h at 25 °C, then weigh each one. Third, weigh 0.6000 g of different dried sample into each 5 mL porcelain crucible, and 6.0000 g of flux into each 25 mL porcelain crucible. Fourth, put the 5 mL porcelain crucibles with weighed samples into a muffle furnace and burn at 1000 °C for 1 h, then take them out and put them into a desiccator to cool for 2 h at 25 °C, then weigh each one. Last, pour each burned sample into an agate mortar to grind until it no longer has the particle sense, then pour in the weighed flux, fully grind and mix, then transfer the mixture to a Pt–Au crucible using spoon and brush. For the stirring rod method, weigh 6.000 g of flux into a cleaned Pt–Au crucible directly, then weigh 0.6000 g of dried sample into a small aluminum dustpan, then pour it into the Pt–Au crucible filled with weighed flux, and then mix them with glass rod manually. For the shaker cup method, first, weigh 6.000 g of flux into a 40 mL commercial shaker cup which is disposable product, and can be used directly. Second, weigh 0.6000 g of dried sample into a small aluminum dustpan, then pour it into the shaker cup filled with flux, then cover the lid. Last, shake it several times manually, and then pour the mixture into a Pt–Au crucible. Next, add two drops of deionized water into the shaker cup, cover the lid, and shake it for several times manually again, then pour the mixture into the Pt–Au crucible to remove powder stuck to the wall.

## 3. Results and Discussion

### 3.1. Operational Process

In quantitative XRF analysis, the whole process can be divided into three steps, which constitute the source of the overall analysis error, namely sampling, sample preparation, and spectrometric analysis [27]. Rudolf Muller stated that the errors due to sample preparation are superimposed onto other errors of measurement, thereby increasing the total error [28]. Furthermore, John et al. reported that the overall error is equal to the square root of the sum of the squares of error of the individual components of the error. Thus, because of the good stability of the XRF instrument and experimental and theoretical correction algorithms available in the software, the largest error stems from sample preparation [29]. This indicates that the sample preparation procedure is the most important step in the analysis, which strongly influences the final quantitative result. To explore the possibility of a new superior sample preparation method, three sample preparation methods were compared in the following aspects: homogeneity of the sample–flux mixture, potential contamination and loss during sample preparation, and sample preparation time.

The homogeneity of the fused glass discs from the three different sample preparation methods was determined by observing the color and mixedness (Figure 1) of the sample–flux mixtures and comparing the elemental mapping images. In the GR method (Figure 1a), the sample and flux were mixed very evenly (consistent mixture color) prior to fusing and had no particle sense. In the ST method (Figure 1b), the sample and flux were mixed unevenly and were in a two-phase separation state. Indeed, the particle point (red arrows) indicated the presence of agglomeration in some of these mixtures. Finally, for the sample–flux mixture prepared by the SH method (Figure 1c), the mixing effect was better than that of the mixture prepared by the ST method, and there was no significant two-phase separation state due to the marked impact between the sample and flux in the shaker cup.

The mixture could be further mixed at the fusing step; thus, we next compared the uniformity of the three different sets of fused glass discs. Figure 2 illustrates the mapping images of some elements of GBW07101. The images indicated that there was no significant difference in the homogeneity of the three methods, and the major and minor elements were very evenly distributed. However, there was a significant difference in the content of volatile elements such as Cl. Indeed, the Cl content in the GR-prepared mixture was significantly lower than that of the mixtures prepared by the other two methods. This was attributed to the muffle furnace burning of the CRMs in the GR method. Notably, although no differences in the uniformity of the fused glass discs were observed in the mapping images, in the actual ST sample preparation process, the fused glass discs often displayed crystallization spots that led to cracking of the fused glass discs (Figure 3). This was mainly attributed to insufficient sample mixing prior to melting and the absence of further decomposition in the fusion machine, which ultimately led to the failure of glass disc formation. Thus, for this method, the mixtures had to be fused two- or three-fold to attain a homogeneous glass disc.

The potential sample contamination and losses during sample preparation were significant for all three preparation methods. The risk of contamination and loss increases with increasing number of sample/flux transfers and thus, the GR method (three transfers) was deemed the most likely to lead to sample contamination and loss, resulting in errors in the analytical results. In addition, in this method, a brush is used to brush the mixture into the Pt–Au crucible, which needs to be cleaned to avoid contamination of the next sample. On the other hand, the ST method requires only one transfer, so that it has the lowest risk of contamination and loss. Contamination during the SH method (two transfers) can be avoided by using a small amount of deionized water, instead of a brush, to transfer any adherent sample. Thus, the risk of contamination and loss of the samples is of the order GR > SH > ST.

Based on our statistical calculations, the sample preparation time for the GR, ST, and SH methods were 6, 2, and 3 min, respectively. GR is the most time-consuming method but simultaneously provides information on the loss on ignition and major and minor elements from one sample. This method can save sample consumption and effectively avoids damage of the Pt–Au crucible by burning of the sample at high temperatures. This is because elements such as C and S are eliminated, while heavy metals such as Pb, Zn, Sn, and Sb are oxidized which, in the absence of burning, would be detrimental to the Pt–Au crucible [30]. ST is the least time-consuming method; however, in this method, the Pt–Au crucible can be easily scratched by the glass rod during mixing. Finally, although slightly longer than that of ST, the SH sample preparation time is still acceptable. Notably, the elements in the rock samples are in oxide form (negligible poisoning), and the damage caused by the unburned samples to the Pt–Au crucible is therefore minimal. Thus, burning of the sample before fusing is not mandatory.

In summary, compared to the GR and ST methods, the SH method is a simpler and more effective sample preparation procedure and can be considered to be a trade-off procedure between uniformity, contamination and loss, and sample preparation time.

### 3.2. LLD

The LLD is defined as the lowest amount of analyte in the sample that can be detected but not necessarily quantitated under the stated experimental conditions [31]. The limit of quantitation (LOQ) is defined as the lowest concentration at which the analyte can be reliably detected and at which some predefined goals for bias and imprecision are met [32]. Notably, according to International Union of Pure and Applied Chemistry (IUPAC) guidelines, the LOQ is three-fold greater than the LLD [33]. In this study, the LLD was calculated from Equation (1) [34]:(1)$$\mathrm{LLD}=\frac{3}{s}\sqrt{\frac{r_{b}}{t_{b}}}$$
where s is the sensitivity (cps/ppm), r**_b_** is the estimated background intensity (cps) at the peak position, and t**_b_** is the total background measurement time (s). The LLDs of the elements, automatically provided by SuperQ software, are listed in Table 2. Generally, the LLD is an important performance characteristic in method validation. The results indicate that the LLD of the fused glass discs produced by the three sample preparation procedures are all adequate for rock samples. No significant differences between the LLD of the SH and ST methods were observed. On the other hand, the LLD of MgO and CaO for these two methods were significantly higher than those of the GR method, while those of Al_2_O_3_ and Na_2_O were significantly lower.

### 3.3. Accuracy and Precision

In this study, 19 CRMs were used to estimate the accuracy and precision of the three different sample preparation procedures for rock samples. The results, including the concentrations and standard deviations of ten major oxides and eight minor elements (WD-XRF), loss on ignition (LOI), and corresponding literature values are presented in Appendix A. The sources of the recommend values of the 19 CRMs are summarized in Appendix A.

In view of the numerous major oxide results (Appendix A), which were not easy to compare, the data were sorted out and mapped as illustrated in Figure 4 and Figure 5. Figure 4 presents the measured values of the major oxides obtained in this study and the recommended values cited from the literature. The solid line represents the regression of the measured results. The overall correlation coefficients of 10 major oxides are summarized in Table 3. This data indicated that the measured values of the different sample preparation procedures all show good correlation with the recommended values. However, it was difficult to determine which procedure was the best because the overall correlation coefficient values of the same element for the different procedures were very close. Thus, other criteria were required for comparison. Figure 5 displays the relative error values of the measured major oxide results. For SiO_2_ and Al_2_O_3_, the relative error values were within 2%, except for RGM-2 which displayed relative errors in the range 2–3%. The points of relative error of SiO_2_ and Al_2_O_3_ for the three sample preparation procedures were on the zero-point line, indicating good accuracy. For TFe_2_O_3_, MgO, CaO, K_2_O, and NaO, the relative errors were within 5%, except for TFe_2_O_3_ in MGL-OShBO and JR-2 and MgO in JG-2, JR-2, and GH, which displayed very large relative errors; CaO in JR-2, AC-E, and GH with relative error values in the range 5–6%; and K_2_O in JGb-2, PM-S, and AN-G with relative error values in the range 5–8%. This occurred because low element concentrations lead to small denominators and thus, a large relative error. The points of relative error of TFe_2_O_3_, CaO, K_2_O, and NaO for the three sample preparation procedures did not differ much, indicating similar accuracies. One exception was MgO with points of relative error from the SH method being significantly closer to the zero-point line than those of the other two methods, thereby indicating better accuracy. In addition, with increasing concentration, the points of relative error of TFe_2_O_3_, MgO, K_2_O, and NaO for the three sample preparation procedures gradually converged to the zero-point line, indicating a gradual improvement in accuracy. Conversely, when the CaO concentration exceeded 12%, the points of relative error deviated from the zero-point line, probably due to the CaO calibration for downward deviation. For TiO_2_, MnO, and P_2_O_5_, the relative errors were within 10%, except for TiO_2_ in JG-2; MnO in JG-2 and RGM-2; and P_2_O_5_ in JG-2, AN-G, GH, JR-2, AC-E, JGb-2, W-2a, and MGL-OShBO. The points of relative error of TiO_2_ and MnO significantly deviated from the zero-point line, indicating poor accuracy; this was attributed to a considerably low concentration. However, with the increase in the element concentration, the points of relative error of these two oxides approached the zero-point line, indicating that the accuracy improved with increasing concentration. Moreover, the TiO_2_ from the SH method was significantly closer to the zero-point line, indicating superior accuracy. The points of relative error of P_2_O_5_ also deviated from the zero-point line when the concentration of P_2_O_5_ was <0.05%. Indeed, the concentration of P_2_O_5_ in JG-2 was lower than the LOQ, indicating that these data are incorrect. The concentrations of P_2_O_5_ in AN-G, GH, JR-2, AC-E, JGb-2, W-2a, and MGL-OShBO were too low so that the accuracy was poor. However, when the P_2_O_5_ concentration was >0.05%, the points were almost on the zero-point line and exhibited good accuracy.

For the minor elements (Appendix A), the data attained were poor. The concentrations of Cr in JG-2, JR-2, RGM-2, AC-E, and GH; Cu in JG-2, JG-1a, JR-2, AC-E, GH, and MGL-OShBO; Ba in JGb-2, JR-2, GH, AN-G, and MGL-OShBO; Ni in JG-2, JR-2, RGM-2, AC-E, and GH; Sr in AC-E; and V in JG-2, JR-2, AC-E, GH, and MGL-OShBO were lower than the LLD, thereby indicating that the data were erroneous. Moreover, the concentrations of Cu in JH-1 and RGM-2, Ba in AC-E, Ni in JG-1a, Sr in JR-2 and GH, and Zn in JG-2 were higher than the LLD but lower than the LOQ, also indicating that the concentrations were inaccurate. In addition, the concentrations of Sr in JG-2 and MGL-OShBO were negative, indicating that the data were erroneous. Thus, the concentrations of Sr in JG-2 and MGL-OShBO were rather low and approached the LOQ, resulting in poor accuracy, while on the other hand, the analytical depth of Sr (3.8 mm) approached the thickness of the fused glass disc (3.6 mm). The analytical depth of fluorescent X-rays is an extremely important parameter in XRF analysis since the fluorescent X-ray intensity increases with increasing thickness up to the analytical depth and remains constant thereafter [19]. Generally, the analytical depth of the specimen for each element can be calculated from its density, average elemental composition, and mass absorption coefficient, and the analytical depths of the fused glass disc for 18 elements calculated in this study are listed in Table 4. Considering that the thickness of a fused glass disc is 3.6 mm, it far exceeds the analytical depth for all major elements and most of the minor elements except for Sr (3.8 mm) and Zr (4.9 mm). Thus, for Sr and Zr, the thickness of the fused glass disc should be fixed to prevent variations in the fluorescent X-ray intensity. Notably, the error can be reduced by weighing constant sample and flux masses, maintaining constant conditions during sample preparation, and using the XRF instrument in rotation mode during measurement. However, the analytical depth of Sr approaches the thickness of the fused glass disc, in which a small change can lead to a significant change in the relative position between the thickness of the fused glass disc and analytical depth, thus affecting the accuracy and precision of the data.

The afforded data revealed that relatively accurate results can be attained with the three sample preparation procedures for samples with concentrations higher than the LOQ. However, the vast data is overwhelming and difficult to compare. Therefore, it is necessary to quote a simple comparative mathematical analysis mode. Laurence Whitty-Léveillé et al. adopted a single value, which considered the accuracy and precision, to successfully compare three types of digestion methods and four types of analytical techniques [35]. Therefore, with reference to the Laurence Whitty-Léveillé method, we adopted two derivate equations from the residual sum of squares to compare the three sample preparation methods. These equations consider the standard deviations associated with the measured and recommended values but do not consider any possible systematic underestimation or overestimation by the different procedures due to the use of the squared error [35,36]:(2)$$J_{S;i}=\frac{{{(x}_{i;tw}{-x}_{i;lit})}^{2}}{\sigma_{i;tw}^{2}{+\sigma }_{i;lit}^{2}}$$
(3)$$J_{S}=\overset{N}{\underset{i=1}{\sum }}J_{S;i}$$

For Equation (2), *x_i;tw_* is the measured value, *x_i;lit_* is the corresponding recommended literature value of *x_i;tw_*, *σ_i;tw_* is the standard deviation, *σ_i;lit_* is the corresponding recommended literature value of *σ_i;tw_*, and *J_S;i_* is the dimensionless criterion value for the elements, which aids the comparison of elements with great absolute concentration variations. Based on Equation (2), accuracy is more important than precision. For Equation (3), N is the number of considered elements and *J_S_* is the dimensionless criterion value for the samples. *J_S_* provides an objective approach to estimate the best sample preparation procedure by summing the accuracy/precision of a given procedure in a single value (*J_S_*). Considering the reproducibility of the measurement, the measured value better approaches the recommended value with decreasing *J_S_*.

Based on the data in Appendix A, the *J_S;i_* values were calculated and listed in Appendix A. The *J_S_* values of 57 CRMs with three sample preparation procedures were also calculated and listed in Table 5.

For the major oxides, the SH method presented the lowest J_S;major_ value (3.95), followed by those attained from the GR (5.57) and ST (5.94) methods. On the other hand, for the minor elements, the ST method presented the lowest J_S;minor_ value (11.36), followed by those attained by the SH (16.64) and GR (33.09) methods. Based on the work of Whitty-Léveillé, the measured values better approach the recommended values with decreasing J_S;major_ and J_S;minor_ values. These data indicate that the GR method is a poor choice for the major oxides and minor elements. If both the major oxides and minor elements are considered, then the ST method displays a lower J_S_ value (16.93) than that of the SH method (20.59). However, notably, the biggest difference between the two methods is from Zr, with a J_SA;Zr_ value of 0.87 for the ST method and 5.39 for the SH method. Thus, if Zr is excluded, the SH method shows a lower J_S;without Zr_ value (15.20) than that of the ST method (16.06). We therefore recommend the use of the J_S;without Zr_ value to evaluate the three methods, whereby the SH method shows the lowest value indicating the best accuracy and precision. As for Zr, the reason for the poor data result by the SH method is yet unknown. We intend to investigate this in the future.

In this paper, the values of the LOI of 19 CRMs are also listed in Appendix A. For JG-2, John Stix et al. have reported a value of 0.41 [37], while the value reported by Antje Herbrich et al. and Roman Golowin et al. was 0.57 [38,39]. The LOI of JG-2 in this study is the same as that reported by John Stix. For JG-1a, John Stix et al. have reported a value of 1.21 [37], while Zongshou Yu et al. have reported a value of 0.55 [40]. The LOI value of JG-1a in this study was 0.68, which approaches that reported by Zongshou Yu. For JA-3, Antje Herbrich et al. and Roman Golowin et al. have reported a value of 0.34 [38,39], John Stix et al. reported a value of 1.08 [37], and Sarah Freund et al. a value of 0.12 [41]. The LOI value of JA-3 in this study was 0.18, which is similar to that reported by Sarah Freund. For JR-2, the LOI value in this study is consistent with that reported by John Stix [37]. For GSP-2, Ingrid Raczek et al. reported a value of 0.96 [42], Zongshou Yu et al. a value of 0.8 [40], and Lindsay J. McHenry a value of 2.88 [43]. The LOI value of GSP-2 in this study was 1.12, which is close to that reported by Ingrid Raczek. For BCR-2, many different values (−0.04, −0.02, −0.01, 0, 0.07, and 0.69) were measured by other authors [40,42,44,45,46,47,48], while the LOI value in this study was −0.06. For BHVO-2, many different values ranging from −0.88 to 0.97 were measured by other authors [40,42,43,44,45,46,47,49,50,51,52,53,54,55], while the LOI in this study was −0.55. For JH-1, the LOI in this study was 2.13, which is higher than the recommended value of 1.78. On the other hand, for JGb-2, AC-E, GH, PM-S, WS-E, AN-G, and MGL-OShBO, the LOI values in this study well agree with the literature recommended values. Finally, the JB-2a, JB-3a, W-2, and RGM-2 LOI values are reported for the first time herein.

## 4. Conclusions

In this study, the advantages and disadvantages of three fusion technique sample preparation procedures for the routine XRF analysis of vast rock samples were compared. In terms of the operational process, the SH method greatly reduced the sample preparation time compared to that of the GR method. It also afforded a more uniform mixture prior to fusing over that attained with the ST method. In terms of the detection limits, the LLD of MgO and CaO attained by the SH method were significantly higher than those attained by the GR method, while those of Al_2_O_3_ and Na_2_O were significantly lower. The LLD of the other tested elements were near identical for the three methods. Additionally, we adopted a dimensionless criterion index, J_S;without Zr_, to evaluate the accuracy and precision. The J_S;without Zr_ value of the SH method was the lowest when Zr was excluded, suggesting that of the three tested methods, this method afforded the most accurate and precise data. Overall, the SH method is highly recommended as a better sample preparation procedure over the other two tested methods.

## Figures and Tables

**Figure 1 sensors-20-05325-f001:**
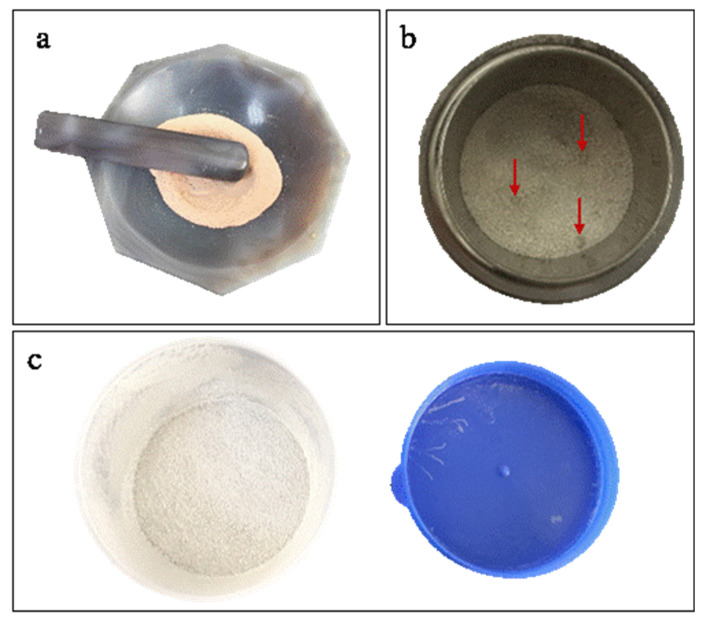
Sample–flux mixtures attained by the (**a**) grinding (GR), (**b**) stirring rod (ST), and (**c**) shaker cup (SH) methods.

**Figure 2 sensors-20-05325-f002:**
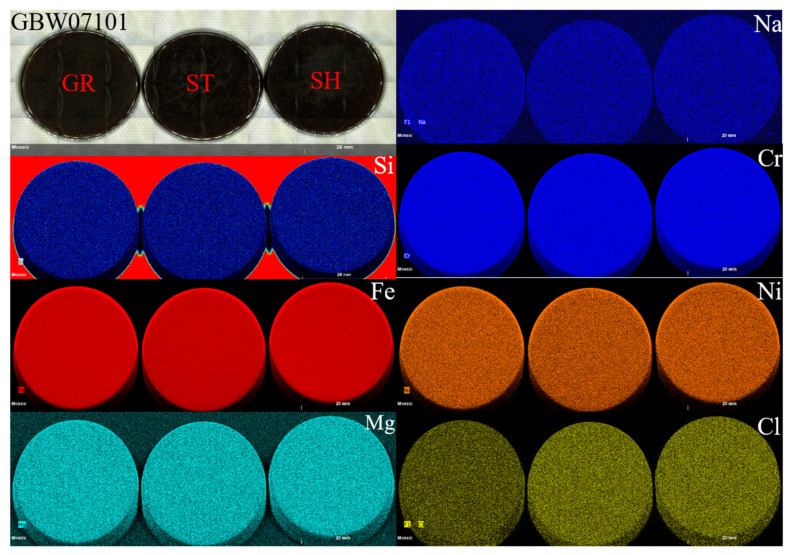
Mapping images of the GBW07101 (ultramafic rock) samples [Scanning area: width, 4645 pixels (92.9 mm) and height, 1615 pixels (32.3 mm); counting time per pixel (20 μm), 5 ms].

**Figure 3 sensors-20-05325-f003:**
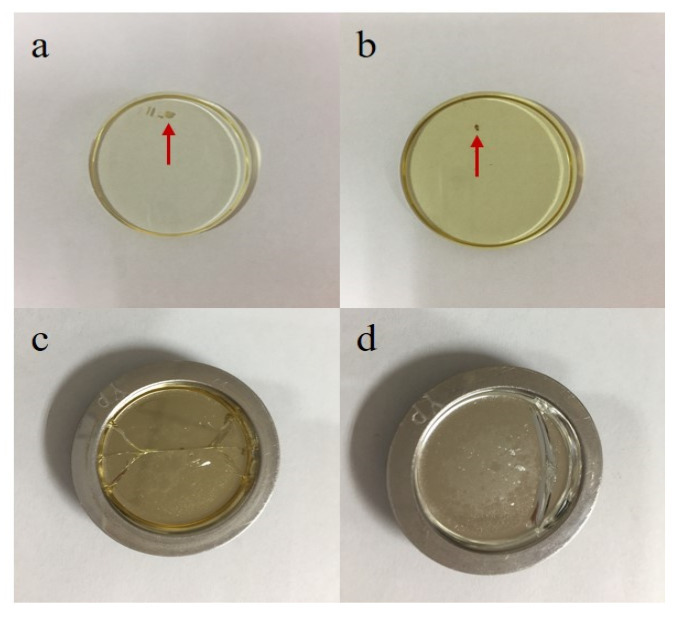
Fused glass discs prepared by the stirring rod (ST) method (letters (**a**) and (**b**) represent the fused glass discs with crystallization spots; letters (**c**) and (**d**) represent the cracked glass discs).

**Figure 4 sensors-20-05325-f004:**
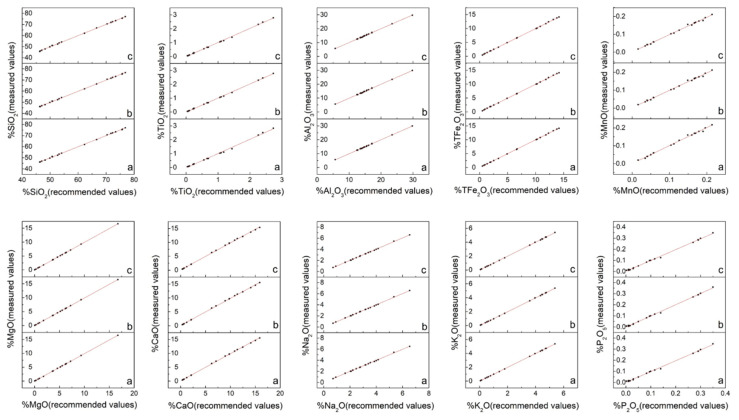
Representation of the measured and recommended values for the major oxides (letters (**a**), (**b**), and (**c**) represent the grinding (GR), stirring rod (ST), and shaker cup (SH) methods, respectively).

**Figure 5 sensors-20-05325-f005:**
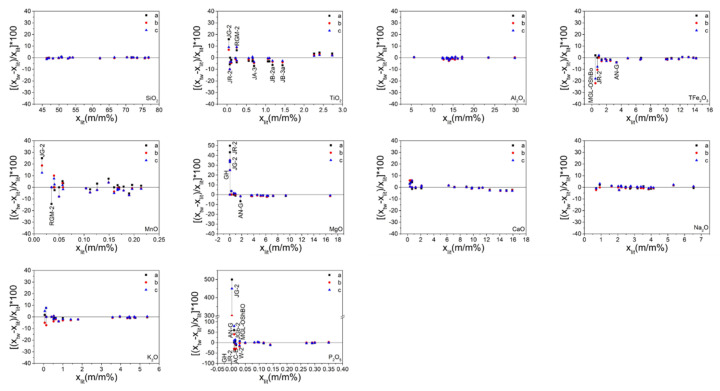
Representation of the relative errors of the measured values for the major elements (letters (**a**), (**b**), and (**c**) represent the grind (GR), stirring rod (ST), and shaker cup (SH) methods, respectively). The values attained from different sample preparation methods, which are of the same sample, are connected in a straight line for clarity.

**Table 1 sensors-20-05325-t001:** Instrumental parameters used in wavelength dispersive X-ray fluorescence spectrometry (WD-XRF).

**Element**		Si	Ti	Al	Fe	Mn	Mg	Ca	Na	K	P	Cr	Cu	Ba	Ni	Sr	V	Zr	Zn
**Line**		Kα	Kα	Kα	Kα	Kα	Kα	Kα	Kα	Kα	Kα	Kα	Kα	Lα	Kα	Kα	Kα	Kα	Kα
**Crystal**		PE 002	LiF 200	PE 002	LiF 200	LiF 200	PX1	LiF 200	PX1	LiF 200	Ge 111	LiF 200	LiF 200	LiF 200	LiF 200	LiF 200	LiF 200	LiF 200	LiF 200
**Detector**		Flow	Flow	Flow	Flow	Flow	Flow	Flow	Flow	Flow	Flow	Flow	Flow	Flow	Flow	Scint.	Flow	Scint.	Scint.
**Voltage/kV**		30	40	30	60	60	30	30	30	30	30	40	60	40	60	60	40	60	60
**Intensity/mA**		120	90	120	60	60	120	120	120	120	120	90	60	90	60	60	90	60	60
**Collimator/μm**		150	300	300	300	300	300	150	700	300	300	300	150	300	150	150	300	150	150
**Counting time/s**		60	30	40	26	36	40	40	40	30	30	40	40	60	40	40	40	40	40
**The** **Grinding Method**	**Peak Angle/2θ**	109.0896	86.1196	144.7838	57.5056	62.9604	23.2312	113.0678	28.0754	136.6656	141.1432	69.3396	45.0206	87.1566	48.6566	25.1452	76.9228	22.5018	41.7962
	**Bg1**	2.2312	1.7800	−1.6878	0.9264	0.9414	2.3962	−0.9160	−2.2264	2.6668	−1.7890	0.9340	0.8988	0.9548	0.6472	−0.5904	−0.7804	−0.8338	0.7076
	**Bg2**			3.1288				1.7958	2.3114		2.5214							0.9642	
	**PHD**	33–67	38–62	32–69	37–63	33–55	30–72	35–65	30–70	35–65	30–56	38–62	36–53	33–53	35–53	30–67	33–53	30–69	29–67
**The Stirring Rod Method**	**Peak Angle/2θ**	109.0896	86.1188	144.7788	57.5038	62.9588	23.2312	113.0678	28.0660	136.6650	141.1414	69.3388	45.0214	87.1556	48.6578	25.1452	76.9202	22.5038	41.7962
	**Bg1**	2.2642	−0.9434	−1.1734	0.9528	1.8912	2.3820	−0.9674	−1.9244	2.4010	−1.7264	1.6960	1.6544	1.4378	0.8044	−0.5490	1.5670	−0.7000	0.8320
	**Bg2**			3.1132				1.7726	2.4482		2.5190							0.7962	
	**PHD**	31–72	36–63	28–72	34–66	33–53	29–75	32–70	23–66	31–69	26–58	37–62	35–52	33–53	34–53	22–78	32–53	24–78	25–70
**The Shaker Cup Method**	**Peak Angle/2θ**	109.0896	86.1226	144.7788	57.5056	63.0042	23.2312	113.0678	28.0708	136.6650	141.1414	69.3388	45.0214	87.1556	48.6566	25.1452	76.9258	22.5084	41.7962
	**Bg1**	2.3066	1.4588	−1.1734	0.9434	1.7556	2.2216	−1.0112	−2.0756	2.6322	−1.6698	1.6750	1.4692	1.4678	0.7302	0.6208	1.7620	−0.5896	0.6358
	**Bg2**			3.1338				1.9038	2.3678		2.7170							0.8426	
	**PHD**	30–71	35–65	22–78	36–64	34–52	29–75	32–69	24–58	31–71	26–60	37–63	36–52	33–53	34–52	22–78	33–53	29–69	26–72

**Table 2 sensors-20-05325-t002:** Data for the range of standard sample compositions and limits of detection (LLD) of the three sample preparation procedures.

Number	Elements	Range of Standard Sample Composition	LLD (μg g^−1^)		
	Major	(m/m% ^a^)	GR	ST	SH
1	SiO_2_	0.62–90.36	157.94	147.90	148.11
2	TiO_2_	0.004–7.69	25.73	29.34	30.55
3	Al_2_O_3_	0.1–59.20	271.70	161.83	159.69
4	TFe_2_O_3_ ^b^	0.075–25.65	17.02	16.28	15.72
5	MnO	0.001–0.43	11.49	11.09	10.85
6	MgO	0.006–49.40	54.70	85.41	85.68
7	CaO	0.04–51.10	35.89	49.68	49.95
8	Na_2_O	0.008–10.59	79.77	77.23	62.53
9	K_2_O	0.003–12.81	13.82	20.37	19.17
10	P_2_O_5_	0.002–6.06	17.20	16.67	14.48
	Minor	(μg g^−1^)			
11	Cr	1.6–15500	11.13	11.19	11.32
12	Cu	0.82–1230	7.73	7.13	7.22
13	Ba	6.4–4000	53.28	52.67	52.28
14	Ni	0.9–3780	5.63	5.52	5.36
15	Sr	2.3–12000	3.82	4.04	4.05
16	V	0.0022–768	12.88	13.16	13.51
17	Zr	0.7–1540	2.06	2.15	1.72
18	Zn	3.5–1300	4.44	5.74	5.78

^a^ m/m% is the mass percentage; ^b^ TFe_2_O_3_ is the total iron oxide as ferric iron.

**Table 3 sensors-20-05325-t003:** Results of the overall correlation coefficients between the measured and recommended values of the major oxides.

Major Oxides	Overall Correlation Coefficient		
	The Grinding (GR) Method	The Stirring Rod (ST) Method	The Shaker Cup (SH) Method
SiO_2_	0.99958	0.99972	0.99980
TiO_2_	0.99710	0.99893	0.99939
Al_2_O_3_	0.99941	0.99930	0.99942
TFe_2_O_3_	0.99987	0.99983	0.99983
MnO	0.99561	0.99649	0.99664
MgO	0.99995	0.99998	0.99999
CaO	0.99967	0.99960	0.99954
Na_2_O	0.99929	0.99951	0.99951
K_2_O	0.99994	0.99992	0.99990
P_2_O_5_	0.99813	0.99856	0.99828

**Table 4 sensors-20-05325-t004:** Analytical depths of the fluorescent X-rays in the fused glass disc.

Analytical Line	Energy/keV	Analytical Depth/μm	Analytical Line	Energy/keV	Analytical Depth/μm
Si Kα	1.74	23	P Kα	2.01	21
Ti Kα	4.51	281	Cr Kα	5.41	530
Al Kα	1.49	10	Cu Kα	8.05	1080
Fe Kα	6.40	560	Ba Lα	4.47	272
Mn Kα	5.90	461	Ni Kα	7.48	916
Mg Kα	1.25	4	Sr Kα	14.17	3831
Ca Kα	3.69	158	V Kα	4.95	430
Na Kα	1.04	3	Zr Kα	15.78	4875
K Kα	3.31	95	Zn Kα	8.64	1279

**Table 5 sensors-20-05325-t005:** Weighted discrepancies between the measured and recommended values of 19 reference materials obtained by the three different sample preparation procedures and analyzed by wavelength dispersive X-ray fluorescence spectrometry (WD-XRF).

Elements	J_ST;i_ ^a^			n ^b^	J_SA;i_ ^c^		
	GR	ST	SH		GR	ST	SH
Major							
SiO_2_	5.54	5.47	4.28	17	0.33	0.32	0.25
TiO_2_	16.96	7.2	4.56	16	1.06	0.45	0.29
Al_2_O_3_	5.44	6.39	3.27	17	0.32	0.38	0.19
TFe_2_O_3_	2.03	16.06	11.6	17	0.12	0.94	0.68
MnO	30.57	5.62	5.97	17	1.80	0.33	0.35
MgO	9.62	8.1	5.3	16	0.60	0.51	0.33
CaO	16.05	23.96	21.59	17	0.94	1.41	1.27
Na_2_O	1.33	1.46	2.53	17	0.08	0.09	0.15
K_2_O	2.91	7.32	2.91	17	0.17	0.43	0.17
P_2_O_5_	8.31	11.43	4.25	16	0.52	0.71	0.27
Minor							
Cr	9.65	3.71	4	12	0.80	0.31	0.33
Cu	87.84	22.48	24.1	9	9.76	2.50	2.68
Ba	2.57	5.24	7.71	12	0.21	0.44	0.64
Ni	11.35	7.72	5.57	10	1.14	0.77	0.56
Sr	55.94	40.44	44.52	12	4.66	3.37	3.71
V	7.6	3.61	4.96	11	0.69	0.33	0.45
Zr	14	14.83	91.66	17	0.82	0.87	5.39
Zn	240.05	44.41	45.96	16	15.00	2.78	2.87
J_S;major_ ^d^					5.94	5.57	3.95
J_S;minor_ ^e^					33.09	11.36	16.64
J_S_ ^f^					39.03	16.93	20.59
J_S;without Zr_ ^g^					38.21	16.06	15.20

^a^ J_ST;i_ is the sum of the J_S;i_ values of an element from different samples; ^b^ n is the total number of the effective measured value of an element; ^c^ J_SA;i_ is the value of J_ST;i_ divided by n; ^d^ J_S;major_ is the sum of the J_SA;i_ values of major oxides; ^e^ J_S;minor_ is the sum of the J_SA;i_ values of minor elements; ^f^ J_S_ is the sum of the J_SA;i_ values of major oxides and minor elements; ^g^ J_S;without Zr_ is the sum of the J_SA;i_ values of elements including SiO_2_, TiO_2_, Al_2_O_3_, TFe_2_O_3_, MnO, MgO, CaO, Na_2_O, K_2_O, P_2_O_5_, Cr, Cu, Ba, Ni, Sr, V, and Zn.

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
