# Peer review of "Comparative Study of Three Mixing Methods in Fusion Technique for Determining Major and Minor Elements Using Wavelength Dispersive X-ray Fluorescence Spectroscopy"

_sensors, 2020, doi:10.3390/s20185325_

Round 1

Reviewer 1 Report

Well written and designed overall. Some of the tables use small fonts which can be difficult to read.

The detail used to outline the procedures and samples used is refreshing.

Line 178: Please clarify why the Cl content would vary. I see that you stated that the burning in the muffle furnace can be the cause but  please elaborate as the the meaning in terms of chemistry.

Line 190: What instrument was used to map the elements in figure 4? also the images are very dark and when printed in black and white are difficult to read.

Line 224: what does cps mean? Also can you define Limit of Quantification as well. You discuss it later but do not define it. Is LLD different than LOD? please clarify.

Line 223: Please ensure that you are following the correct process for displaying the proper number of significant figures for your data. This will be true for all of your tabulated data.

Line 252: Cannot read the fonts for figure 6. Also would error bars on this data be helpful in discussing the overall uncertainty between the different techniques?

Line 255: Cannot read the fonts for figure 7. 

Reviewer 2 Report

This manuscript includes the analysis by XRF to the some element and chemicals. Some tables and figures are better to change. the comments are as follows,

1) to Table 2, please shorten, move to supplement data or delete. 

2) please delete Figures 1 and 2. Tables 3 and 4 are rewritten in text.

3) please delete table 6.

4) Figure 6 is better to include partially the reference data.

5) in introduction, the heterogeneity of the prepared sample has a problem for the distribution. Can you evaluate the heterogeneity chemically?

Round 2

Reviewer 2 Report

The manuscript is well revised according to the reviewer's comments. But actually, the manuscript is not the presentation of ppt. Thus Figures 1-3 should be deleted.

Author Response

Thank you for your comments. Figures 1 and 2 have been relabeled as Figures S1 and S2, respectively, and moved to the supplementary materials. Moreover, the procedure have been rewritten in text. So Tables S2 and S3 are deleted. The other tables and figures in the article have been renumbered. (please see Page 5, Lines 132-162)